# Mechanical Performance of 3D Printed Concrete in Steam Curing Conditions

**DOI:** 10.3390/ma15082864

**Published:** 2022-04-13

**Authors:** Bolin Wang, Xiaofei Yao, Min Yang, Runhong Zhang, Jizhuo Huang, Xiangyu Wang, Zhejun Dong, Hongyu Zhao

**Affiliations:** 1CCCC First Highway Consultants Co., Ltd., Xi’an 710075, China; wbl1993lang@163.com (B.W.); slbyxf@163.com (X.Y.); 2Institute for Smart City of Chongqing University in Liyang, Chongqing University, Liyang 213300, China; zhangrh@cqu.edu.cn; 3College of Aerospace Engineering, Chongqing University, Chongqing 400045, China; 4College of Civil Engineering, Fuzhou University, 2 Xue Yuan Rd., University Town, Fuzhou 350116, China; 5School of Design and Built Environment, Curtin University, Perth 6102, Australia; xiangyu.wang@curtin.edu.au; 6School of Civil Engineering, Suzhou University of Science and Technology, Suzhou 215009, China; knight.cc.ccc@gmail.com; 7School of Civil Engineering, Chongqing University, Chongqing 400045, China

**Keywords:** 3D concrete printing, curing conditions, mechanical capacity, solid waste, anisotropy, sustainability

## Abstract

Three-dimensional (3D) concrete printing (3DCP) technology attracts significant attention from research and industry. Moreover, adequate mechanical performance is one of the primary properties for materials, meeting the demand of structural safety using 3DCP technology. However, research on curing conditions as the significant influence factor of mechanical capacity is required to accelerate the practical application of 3DCP technology. This study aims to explore the impact of various steam curing conditions (heating rate, constant temperature time, and constant temperature) on the mechanical performance of printed concrete containing solid wastes. Moreover, the optimal steam curing conditions are obtained for compressive, tensile, and flexural properties in different directions. Subsequently, anisotropies in the mechanical properties of printed composites and interlayer bonding behaviors are investigated when various curing conditions are employed. The result shows that steam curing conditions and solid waste incorporation improves the interlayer bond for 3D printed cement-based composites.

## 1. Introduction

Additive manufacturing is a method exhibiting the characteristics of saving material, being economical, having high construction speeds, and enabling flexible design, and has been employed in various fields [1,2,3]. In the last decade, the interest in additive manufacturing from the building industry has significantly increased [4,5,6]. The rapid growth and elaborate research in this field indicate that additive manufactured construction structures are not a dream but will be a reality in the near future, although they are still in their nascent stage [7,8,9]. Meanwhile, additive manufacturing in construction has developed varying technologies such as contour crafting, D-shape, and extrusion-based 3DCP [10,11,12]. 3DCP technology is the cementitious composites deposited in layer-by-layer shapes by the coordination of a computer design and the help of a printer [2,13]. Advantages of 3DCP include no requirement for formwork, waste reduction, and significant saving in costs and labor [14,15,16,17,18]. Consequently, 3DCP has attracted the most attention from both academia and industry in the additive manufacturing area [19,20,21].

To ensure structural safety using 3DCP technology, desirable materials using the 3DCP method demand adequate mechanical properties for load bearing [1,19,22]. Additionally, high early strength is essential to ensure the maintenance of printed shapes and printed layers without collapse [23,24,25]. The curing conditions and material mix proportions are the main influencing factors on the mechanical properties of a material [26,27,28]. Moreover, the steam curing method improves the hydration and early strength of materials to benefit 3DCP [29,30,31]. However, improper steam curing technology used in printed concrete creates an interface transition zone between the cementitious material and aggregate resulting in thermal expansion deformation which produces microcracks [32,33]. As the microcracks grow, the long-term performance of steam-cured concrete may be reduced. Furthermore, conventional concrete by 3DCP methodology generally uses fine aggregates, leading to more cement usage, CO_2_ emissions, and energy consumption than traditional casting concrete due to the higher cement content [34,35,36]. Compared to cast concrete, higher evaporation rates, surface drying, and cement content for printed concrete can induce shrinkage cracking due to the lack of formwork [37,38,39]. Disposing of anisotropy (the distinct property of 3D printing materials) is crucial to promoting the use of 3DCP technology in large-scale construction [40,41,42]. Therefore, research on the optimum steam curing conditions of mechanical capacity and anisotropy of 3DCP is required.

The rapid development of industrial production and demolition has led to a huge amount of various solid wastes [43,44,45]. Meanwhile, more than 95% of urban solid wastes are directed to landfill, thus impacting the health of humans and the environment [46,47,48]. As a result, effectively improving the reuse of solid wastes is urgent to obtain sustainability [49,50,51]. FA, SF, and GGBS are solid wastes in the by-product of the energy and smelting industry, which need to be resolved at a considerable cost [52,53,54]. However, cementitious composites using 3DCP technology employ the combination of FA, SF, and GGBS to replace cement, improving sustainability, shrinkage resistance, steam curing adaptability, durability, and the mechanical properties of materials [25,55,56]. Printed materials with waste minerals incorporated improve the internal pore structure and the particle fineness gradation, based on the theory of dense packing [57,58,59].

This study aims to investigate the influence of mechanical performance when various steam curing conditions are employed on the concrete using 3DCP technology. Orthogonal experiments are conducted to reduce the experimental workload and obtain a high-sensitivity evaluation indicator. Additionally, a coefficient is used to quantitatively assess the anisotropy of the materials. Subsequently, a further analysis of printed interlayer bonding performance is conducted. Thereby, the optimum printed concrete steam curing conditions for various mechanical performance and anisotropy are acquired.

## 2. Materials and Methods

### 2.1. Raw Materials

The precursor materials of the samples in this study were 42.5 R Portland cement (produced by China Anhui Conch Group Company Limited, Wuhu, China), FA (Class F, obtained from Shenglong Technology Industry Co., Ltd., Weinan, China), GGBS (produced by Delong Powder Engineering Material Co., Ltd., Xi’an, China), and SF (provided by Linyuan micro silica powder Co., Ltd., Xi’an, China). The chemical compositions of the precursor materials are shown in Table 1. Quartz sand with a particle grading range of between 0.21 mm and 0.55 mm wa used because it has less mud content than natural sand, as shown in Figure 1. Additionally, the good gradation of quartz sand guaranteed that the pipe and nozzle of the printer were not blocked. The physical and mechanical properties of PVA fibers utilized in this study are shown in Table 2. Additionally, the high-efficiency polycarboxylic acid type water reducer and hydroxypropyl methylcellulose (viscosity at 40,000 Cp) were employed to improve the working performance of 3D printed concrete.

### 2.2. Orthogonal Experimental Design of Mix Proportion

The mixing proportions of 3D printed concrete mainly include a water binder ratio, sand binder ratio, polycarboxylate superplasticizer, PVA fiber, hydroxypropyl methylcellulose, etc. The water binder ratio of this experiment was 0.36 based on a trial test considering the balance between printable fluidity and mechanical strength. The sand binder ratio at 1.3 was verified to provide smooth extrusion and reasonable cost. The optimal material mix proportion was determined by previous tests of mechanical and printable capacity [60]. Therefore, the components of FA, SF, and GGBS were 20 wt.%, 15 wt.%, and 10 wt.%, respectively, in the cementitious materials which were calculated when the total percentage of binder materials was 100 wt.%, as shown in Table 3. PVA fiber, hydroxypropyl methylcellulose, and polycarboxylate superplasticizer content in the mortar were 0.21 wt.%, 0.23 wt.%, and 0.016 wt.%, respectively. Hydroxypropyl methylcellulose promotes the dispersion of mortar and fiber, thus enhancing the water-retaining property of the material. Moreover, the addition of hydroxypropyl methylcellulose improved viscosity.

The orthogonal test aimed to explore the mechanical capacity of the printed concrete influenced by various steam curing conditions. With reference to Chinese criteria [61] and previous research [62], three factors (heating rate, holding time, holding temperature) and three levels were designed in the orthogonal experimental shown in Table 4. The study utilized range analysis because of higher intelligibility and more convenient calculation compared to analysis of variance. The range analysis method utilizes the average value calculation of each level and obtains the range value, and subtracts the minimum level average value from the maximum level average value. The primary and secondary relationships affecting the index value are determined by the range value.

### 2.3. Load Direction Definition

Load direction was determined to explore the anisotropy of printed composites. Figure 2a shows that the three directions of X, Y, and Z are orthogonal. The X, Y, and Z directions are the outer points perpendicular to the centre of the cross-section of the printed layer, the profile center of the printed layer, and the center of the plane section of the print layer, respectively. F_xy_ and F_xz_ are the X-direction line loads, which extend in Y direction and Z direction, respectively, as shown in Figure 2b. F_yx_ and F_yz_ represent the line loads in the Y direction extending in the X direction and Z direction, respectively. F_zx_ and F_zy_ denote the Z-direction line loads extending in the X direction and Y direction, respectively.

### 2.4. Mechanical Performance Test

The mix proportion of printable mortar referred to in Table 3 was used for the mechanical performance test. The structure was extruded and deposited by 3DCP equipment according to the designative printing path, as shown in Figure 3. Additionally, the printing direction of each layer was Y. The height, width of printing nozzle, and nozzle moving velocity were 38 mm, 14 mm, and 12 cm/s, respectively. To improve printing efficiency and structure quality, a large-sized printing nozzle was used for the structure fabrication. Moreover, it enabled a more stable deposition of the printing filaments compared to the circular nozzle, preventing collapse. Each layer was manufactured with 12 round-trip continuous paths and the number of vertical deposition layers was 20. The printing path interval was 40 mm and the height of each layer was set to 13 mm. The structure used a bucket (0.05 m^3^) of printing material and its printing time was 15 min. The open time of the mortar was 40 min. The printing and equipment parameters were based on previous experimental results of buildability and extrudability [60]. The round-trip continuous paths ensured that the mutual extrusion of printing filament was not improved.

By adopting the Chinese national standard GB/T 50081-2002 [63,64], compressive, flexural, and tensile tests were employed to evaluate the mechanical performance influenced by various steam curing conditions. The cube samples (100 mm × 100 mm × 100 mm) and prismatic specimens (50 mm × 50 mm × 200 mm), used in different direction tests, were obtained by structural cutting after 8 h of natural maintenance, as shown in Figure 4a,b. A printed structure was cut into three cube compression test samples; and three cube specimens were used in the X, Y, and Z directions splitting tension test, as shown in Figure 4a. The splitting tension test specimens in the X direction were loaded in an XY or XZ direction, specimens in the Y directions were loaded in an YX or YZ direction, and specimens in the Z directions were loaded in an ZX or ZY direction. Six prismatic flexure tests specimens loaded in various directions (XY, XZ, YX, YZ, ZX, and ZY) were acquired from the printed structure. To ensure the accuracy of the mechanical properties experiment, the cube or prismatic specimens, which were employed in the same direction test, were cut from the same position in the printed structures and had the same interlayer number. After an 8-h cut time, the test samples were placed into a zky-400b steam box to cure one circle, as shown in Figure 4c.The steam curing conditions of the printed samples refer to the orthogonal experimental design (Table 4). Subsequently, the test samples were cured for 28 days in natural conditions. The experiments of 28-day compressive, flexural, and tensile performance utilized 9 group samples with various steam curing conditions and 2 control groups cured in natural conditions. Subsequently, uniaxial compressive strength in the X, Y, and Z directions was determined by compression tests in which each group contained three cube specimens, as shown in Figure 5a.Each group of flexural and splitting tensile capacity tests contained six prismatic specimens to be under the line flexural and tensile load of XY, XZ, YX, YZ, ZX, and ZY directions, as shown in Figure 5b,c. Moreover, the average and coefficient of variation of mechanical performance were obtained by measuring three group samples at each curing condition from Table 4. 

### 2.5. Anisotropy Assessment

This study applied an anisotropy coefficient to represent the influence of the printing process on the mechanical behavior of the materials, as described in Equations (1) and (2).
(1)$$f_{avg}=\frac{\sum_{n=1}^{i}f_{xn}+\sum_{n=1}^{i}f_{yn}+\sum_{n=1}^{i}f_{zn}}{3i}$$
(2)$$I_{a}=\frac{\sqrt{{\left(f_{x}{}_{1}-f_{avg}\right)}^{2}+\ldots +{\left(f_{x}{}_{i}-f_{avg}\right)}^{2}+{\left(f_{y}{}_{1}-f_{avg}\right)}^{2}+\ldots +{\left(f_{y}{}_{i}-f_{avg}\right)}^{2}+{\left(f_{z}{}_{1}-f_{avg}\right)}^{2}+\ldots +{\left(f_{z}{}_{i}-f_{avg}\right)}^{2}}}{f_{avg}}$$
where *i* is the number of load directions in the direction of the main load, *f_xi_*, *f_yi_*, and *f_zi,_* are the average strength of *i*-th load direction in the X, Y, and Z direction, respectively, *f_avg_* is the average strength of all loads, and *I_a_* is the anisotropy coefficient. Thereby, the value of *I_a_* is the positive correlation with anisotropy of the printed material. The smaller value of *I_a_* ought to be employed to improve the feasibility of printed structures when the requirement for mechanical property directions of the structure are not determined.

## 3. Results and Discussion

### 3.1. Analysis of Mechanical Performance in Various Curing Conditions

#### 3.1.1. Compressive Performance

The compressive strength (CS) test data and variation coefficient of the samples in the various curing conditions were summarized in the L9 (3^3^) orthogonal experimental table (Table 5). The average values of CS_x_, CS_y_, and CS_z_ were obtained by the results and calculations of the orthogonal test, as shown in Figure 6. The CS_x_ range values in the various curing conditions of heating-up rate, thermostatic period, and thermostatic temperature were 7.70 MPa, 1.90 MPa, and 3.83 MPa, respectively, and determined the degree of influence of the three steam curing conditions (heating-up rate > thermostatic temperature > thermostatic period). Meanwhile, the CS_y_ range values in the varying conditions of heating-up rate, thermostatic period, and thermostatic temperature were 4.23 MPa, 7.50 MPa, and 5.47 MPa, respectively, which determined the influence degree of thermostatic period > thermostatic temperature > heating-up rate. Furthermore, the CS_z_ range values in the various conditions of heating-up rate, thermostatic period, and thermostatic temperature were 4.37 MPa, 8.23 MPa, and 7.30 MPa, respectively, which determined the influence degree of thermostatic period > thermostatic temperature > heating-up rate.

Figure 6a shows that the CS_x_ had a negative correlation, positive correlation, and parabolic correlation with the curing conditions of heating-up rate, thermostatic period, and thermostatic temperature, respectively. Additionally, Figure 6b shows that negative correlation, parabolic correlation, parabolic correlation were achieved by the CS_y_ with the curing conditions of heating-up rate, thermostatic period, and thermostatic temperature, respectively. Figure 6c shows that CS_z_ had a parabolic correlation with the curing conditions of heating-up rate, thermostatic period, and thermostatic temperature. Moreover, the relation between CS_z_ and heating-up rate was a concave-shape curve, which differed from thermostatic period and thermostatic temperature. Thereby, the optimal parameter of steam curing conditions for CS_x_ and CS_y_ are 10 °C/h heating-up rate, 10h thermostatic period, and 70 °C thermostatic temperature based on the correlation with the various curing conditions. Moreover, the optimal parametersfor steam curing conditions for CS_z_ were 10 °C/h heating-up rate, 8h thermostatic period, and 60 °C thermostatic temperature.

#### 3.1.2. Splitting Tensile Performance

The test data and variation coefficient of the splitting tensile strength (TS) for this study are summarized in the L_9_ (3^3^) orthogonal experimental tables (Table 6 and Table 7). TS of the printed specimen was influenced by the steam curing technology and interlayer effect, as shown in Table 6. Applying steam curing technology to cast samples has some negative effects on the mechanical properties due to a heterogeneous distribution of air voids in the specimens. The cast concrete demonstrates uneven expansion under heating-up conditions due to the thermal expansion of air and water. However, the steam curing technology can improve the TS of printed samples because of an improvement in the bond strength between layers. Additionally, the air voids in the printed samples are relatively homogeneous due to the printing path impact. Small dimensions of the air bubbles and a tight internal structure are produced as a result of the deposition effects of the 3DCP technology. The addition of various solid wastes can improve the hydration of the products and the pore structure, thereby enhancing the steam curing adaptability of 3D printed cement-based materials.

The average values of TS_xy_, TS_xz_, TS_yx_, TS_yz_, TS_zx_, and TS_zy_ are obtained by the results and calculations of the orthogonal test, as shown in Figure 7. The relation between TS_xy_ and the three curing conditions (thermostatic period, thermostatic temperature, heating-up rate) was a parabolic correlation as presented in Figure 7a. Meanwhile, the optimal curing conditions were 15 °C/h (heating-up rate), 8 h (thermostatic period), and 60 °C (thermostatic temperature) for TS_xy_. Figure 7b demonstrates that TS_xz_ had a negative correlation, negative correlation, and parabolic correlation with heating-up rate, thermostatic period, and thermostatic temperature, respectively. A heating-up rate of 10 °C/h, 6h thermostatic period, and 70 °C thermostatic temperature were the optimal curing conditions for TS_xz_. Figure 7c shows that the optimal curing conditions of heating-up rate, thermostatic period, and thermostatic temperature for TS_yx_ were 10 °C/h, 6 h, 50 °C, respectively. Parabolic correlation, negative correlation, and negative correlation, respectively, are the relation between TS_yx_ and the three curing conditions (heating-up rate, thermostatic period, and thermostatic temperature). Figure 7d shows that TS_yz_ had a negative correlation with heating-up rate, negative correlation with thermostatic period, and parabolic correlation with thermostatic temperature. A 15 °C/h heating-up rate, 10h thermostatic period, and 60 °C thermostatic temperature were the optimal curing conditions for TS_yz_. A negative correlation, parabolic correlation, and positive correlation were the relation of TS_yx_ with the heating-up rate, thermostatic period, and thermostatic temperature, respectively, as shown in Figure 7e. For TS_yx_ 10 °C/h, 8 h, and 70 °C were the optimal heating-up rate, thermostatic period, and thermostatic temperature for curing conditions. Figure 7f demonstrates that heating-up rate, thermostatic period, and thermostatic temperature had a negative correlation, parabolic correlation, and positive correlation, respectively, with TS_zy_. Furthermore, the optimal curing conditions were 10 °C/h (heating-up rate), 8 h (thermostatic period), and 70 °C (thermostatic temperature) for TS_zy_.

#### 3.1.3. Flexural Performance

The L_9_ (3^3^) orthogonal experimental tables (Table 8 and Table 9) summarize the test data and variation coefficients of flexural strength (FS). Figure 8 shows that the average values of FS_xy_, FS_xz_, FS_yx_, FS_yz_, FS_zx_, and FS_zy_ were obtained by the results and calculations of the orthogonal test. Compared to the control cast, the flexural strength of the printed samples was commonly decreased because of the weak surface of the printing interlayer. Besides, the practical structure ought to utilize ZY and YZ directions to resist the flexural load.

The optimal curing conditions were 15 °C/h (heating-up rate), 10 h (thermostatic period), and 60 °C (thermostatic temperature) for FS_xy_, as presented in Figure 8a. Figure 8b demonstrates the optimal steam curing conditions (15 °C/h heating-up rate, 10 h thermostatic period, and 50 °C thermostatic temperature) for FS_xz_. Figure 8c shows that the optimal curing conditions of heating-up rate, thermostatic period, and thermostatic temperature for FS_yx_ were 10 °C/h, 10 h, and 70 °C, respectively. Figure 8d exhibits that 20 °C/h heating-up rate, 8h thermostatic period, 50 °C thermostatic temperature were the optimal curing conditions for FS_yz_. Dor FS_yx,_ 20 °C/h, 8 h, and 60 °C were the optimal heating-up rate, thermostatic period, and thermostatic temperature of curing condition, as shown in Figure 8e. Figure 8f demonstrates that the optimal curing conditions were 10 °C/h (heating-up rate), 8 h (thermostatic period), and 70 °C (thermostatic temperature) for FS_zy_.

### 3.2. Anisotropy Assessment in Various Curing Conditions

The test data of compressive strength, splitting tensile strength, and flexural strength employed Equations (1) and (2) to obtain the results of various anisotropy coefficients, as demonstrated in Table 10. The table shows that the anisotropy coefficient of splitting tensile performance was similar to the anisotropy coefficient of flexural performance and two-time compressive performance. The improvement of the weak interlayer surface of the printed concrete with added 0.21 wt% PVA fiber was limited by the brittleness of cementitious material. However, the steam curing condition improved the hydration of the materials, thus accelerating the development of composite strengths.

#### 3.2.1. Anisotropy Assessment of Compressive Performance

The average anisotropy coefficient of compressive strength is shown in Table 10 and calculated with various curing conditions (heating-up rate, thermostatic period, and thermostatic temperature), as shown in Figure 9. The compressive strength range values of various heating-up rates, thermostatic periods, and thermostatic temperatures were 3.34%, 17.20%, and 14.83%, respectively, obtaining the degree of influence of the three curing conditions on I_ac_ (thermostatic period > thermostatic temperature > heating-up rate). Figure 9 shows the relations between compressive strength and the three curing conditions (heating-up rate, thermostatic period, and thermostatic temperature) had a parabolic correlation. The influence of thermostatic period and thermostatic temperature exhibited the same variation trend. Moreover, the I_ac_ variation values with the influence of the thermostatic period and thermostatic temperature were similar. With the curing time increased, the influence degrees of the three curing conditions were reduced for I_ac_. Thereby, the printed samples possessed the lowest anisotropy of compressive performance when heating-up rate, thermostatic period, and thermostatic temperature were 20 °C/h, 8 h, and 60 °C, respectively.

#### 3.2.2. Anisotropy Assessment of Splitting Tensile Performance

Figure 10 shows that the average anisotropy coefficient of the splitting tensile strength was calculated by the data of the orthogonal test (Table 10). Furthermore, the I_ap_ range values of varying curing conditions of heating-up rate, thermostatic period, and thermostatic temperature were 6.75%, 10.59%, and 22.77%, respectively, obtaining the influence degree of thermostatic temperature > thermostatic period > heating-up rate. The I_ap_ had the positive correlation, parabola correlation, and negative correlation with the curing conditions of heating-up rate, thermostatic period, and thermostatic temperature, respectively. Hence, 20 °C/h heating-up rate, 8h thermostatic period, and 70 °C thermostatic temperature were the optimum steam curing condition to reduce the anisotropy of splitting tensile performance.

#### 3.2.3. Anisotropy Assessment of Flexural Performance

The data of the orthogonal test in Table 10 was used to calculate the average anisotropy coefficient of flexural strength, as shown in Figure 11. The I_az_ strength range values of the various curing conditions of heating-up rate, thermostatic period, and thermostatic temperature were 9.52%, 6.56%, and 9.91%, respectively, obtaining the influence degree of three solid wastes on I_az_ (thermostatic temperature > heating-up rate > thermostatic period). Simultaneously, the I_az_ had a parabola correlation with the three steam curing conditions (heating-up rate, thermostatic period, and thermostatic temperature). Therefore, the lowest anisotropy of flexural performance for printed concrete was acquired when heating-up rate, thermostatic period, and thermostatic temperature were 15 °C/h, 10 h, and 60 °C, respectively.

### 3.3. Analysis of Interlayer Bonding Capacity

Figure 12a shows that the sequence of compressive performance averages were CS_y_ > CS_x_ > CS_z_. The compressive performance averages of the printed samples were higher than the control cast. Figure 12b,c demonstrates that the splitting tensile and flexural performance averages of the printed samples were lower than the cast. However, the splitting tensile performance averages of the printed specimens in the XZ and ZX directions were close to cast. For the flexural capacity, the averages of the printed specimens in the YZ and ZY directions were 0.52 MPa and 0.6 MPa, slightly lower than cast, respectively.

The interlayer bonding capacity influences the mechanical performance of printed concrete and is related to the steam curing method, solid waste incorporation, and the printing parameters. The impact of steam curing conditions on mechanical performance is discussed in Section 3.1 and Section 3.2. For solid waste, printed material incorporates various solid wastes to improve the microcrystalline nuclear effect, pozzolan effect, and micro aggregate effect leading to a dense structure. The microcrystalline nuclear effect accelerates the hydration reaction and improves the homogenization of the hydration product distribution. Furthermore, solid waste absorbing the calcium hydroxide of hydration enhances the hydration of the cement to produce more C-S-H gels, thereby improving the microstructure of the material. Meanwhile, the void ratio reduction of the material enables the cohesion of the aggregate interface up and increases the weak interface of printed concrete.

The printable filament height and weight were 38mm and 13mm, respectively. The layer height of printable mortar was lower than the nozzle height of the printer due to the additional pressure of the upper printed layer. Figure 13a,b exhibit the diagrammatic sketches of horizontal interlayer and vertical interlayer, respectively, for the printed materials using the 3DCP method. Figure 13c and Figure 14 show that the main crack angle generated on the C plane was oblique 45°, which was different from cast (20°–30°) when the X-direction compression was loaded on the samples. Additionally, some vertical cracks were generated on the C plane because the weak interlayer influenced the slenderness ratio and shear span ratio of the printed material to result in shearing failure.

Figure 15 demonstrates that a cross-shaped main crack and a horizontal main crack were generated on the C plane and the reverse plane, respectively, when the Y-direction compression was loaded on the samples. Moreover, a slide was created on the weakest vertical interlayer of the printed specimen when the compression was loaded, hence proving that the horizontal interlayer bond was higher than the vertical interlayer bond. Two main vertical cracks and shear oblique cracks were generated on the forward and reverse plane respectively of the A plane when the Z-direction compression was loaded on the samples, as shown in Figure 16. Due to the rectangular printable filament, some controllable deformation may improve the vertical interlayer bond and dense microstructure at the deposition effect, thereby reducing anisotropy for material. Hence, the improvement of fluidity is beneficial to the horizontal interlayer bond of materials without horizontal additional pressure.

## 4. Conclusions

This study proposed that the anisotropy coefficient and orthogonal experiment be applied to evaluate the mechanical performance and anisotropy of concrete using 3DCP technology when various steam curing conditions are used for printed materials. Subsequently, the optimum steam curing conditions and interlayer bonding were investigated. The following conclusions can be drawn:The most influential factors for solid wastes for CS_x_, CS_y_, and CS_z_ are heating-up rate, thermostatic period, and thermostatic period, respectively. The optimal steam curing conditions of CS_x_ and CS_y_ are 10 °C/h heating-up rate, 10 h thermostatic period, and 70 °C for the thermostatic temperature. The optimal F_z_ curing conditions are 10 °C/h heating-up rate, 8h thermostatic period, and 60 °C thermostatic temperature.The optimal curing conditions (thermostatic period, thermostatic temperature, heating-up rate) are 15 °C/h, 8 h, and 60 °C for TS_xy_; 10 °C/h, 6 h, and 70 °C for TS_xz_; 10 °C/h, 6 h, and 50 °C for TS_yx_; 15 °C/h, 10 h, and 60 °C for TS_yz_; 10 °C/h, 8 h, and 70 °C for TS_zx_; 10 °C/h, 8 h, and 70 °C for TS_zy_; respectively.The optimal flexural strengths are obtained when the thermostatic period, thermostatic temperature, heating-up rate are 15 °C/h, 10 h, 60 °C for FS_xy_; 15 °C/h, 10 h, 50 °C for FS_xz_; 10 °C/h, 10 h, 70 °C for FS_yx_; 20 °C/h, 8 h, 50 °C for FS_yz_; 20 °C/h, 8 h, 60 °C for FS_zx_; and 10 °C/h, 8 h, 70 °C for FS_zy_; respectively.The optimal heating-up rate, thermostatic period, and thermostatic temperature are 20 °C/h, 8 h, 60 °C for compressive performance; and 20 °C/h, 8 h, 70 °C for splitting tensile performance; and 15 °C/h, 10 h, 60 °C for flexural performance; respectively, to reduce the anisotropy.The interlayer bonding capacity is influenced by steam curing conditions, solid waste incorporation, and printing parameters. Solid waste incorporation can improve the microstructure and interface bond of the printed concrete. Furthermore, some controllable deformation may improve the vertical interlayer bond and dense microstructure at the deposition effect, thereby reducing anisotropy for the material. Meanwhile, the bond of the horizontal interlayer mainly depends on the fluidity of materials without horizontal additional pressure.

## Figures and Tables

**Figure 1 materials-15-02864-f001:**
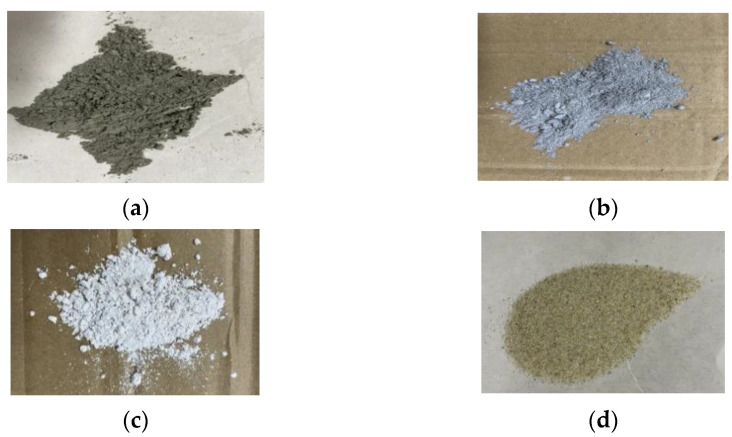
(**a**) FA, (**b**) SF, (**c**) GGBS, and (**d**) quartz sand used in specimens of this experiment [60].

**Figure 2 materials-15-02864-f002:**
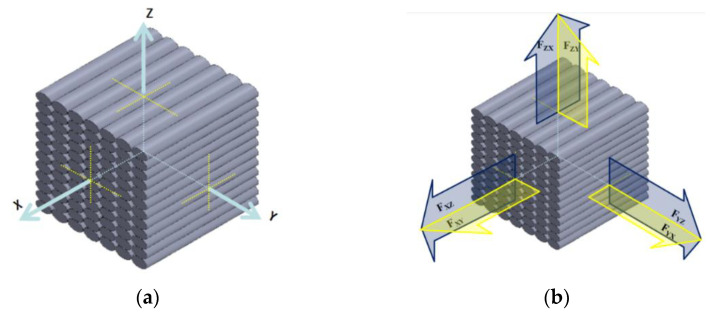
(**a**) Coordinate system, (**b**) loading direction.

**Figure 3 materials-15-02864-f003:**
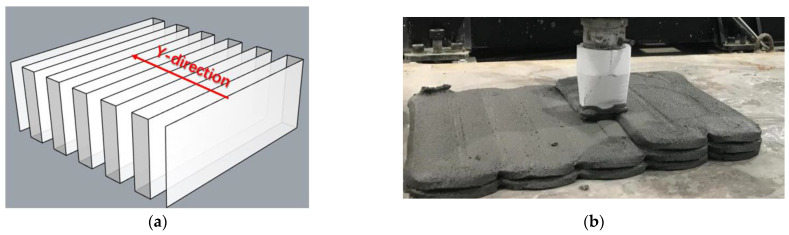
(**a**) Printing path model, (**b**) printing process [60].

**Figure 4 materials-15-02864-f004:**
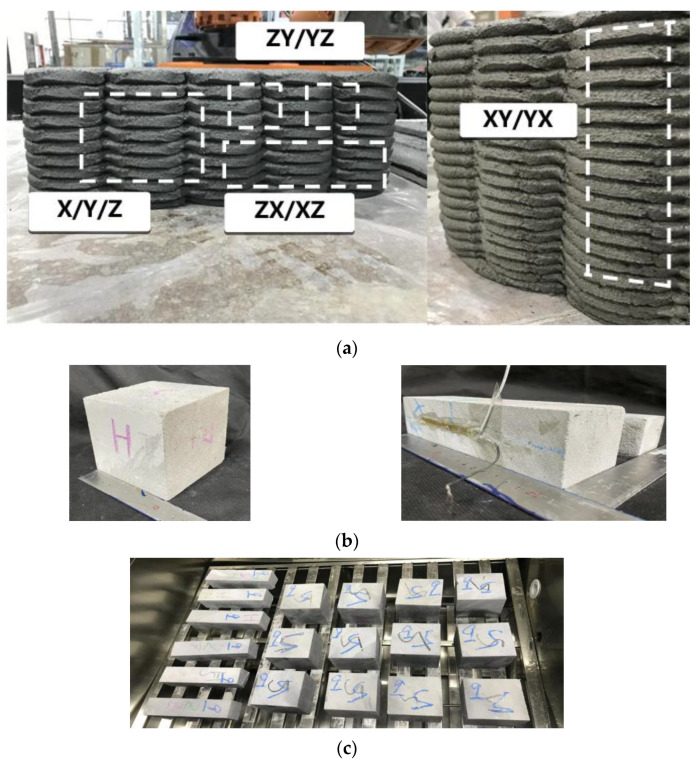
(**a**) Cutting position, (**b**) specimen size, and (**c**) steam curing condition [60].

**Figure 5 materials-15-02864-f005:**
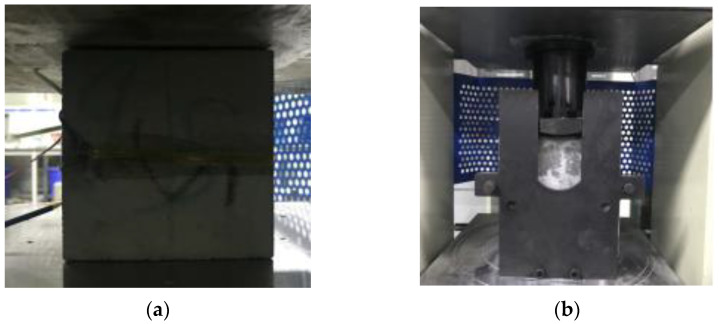
(**a**) Compressive, (**b**) splitting tensile capacity, and (**c**) flexural tests.

**Figure 6 materials-15-02864-f006:**
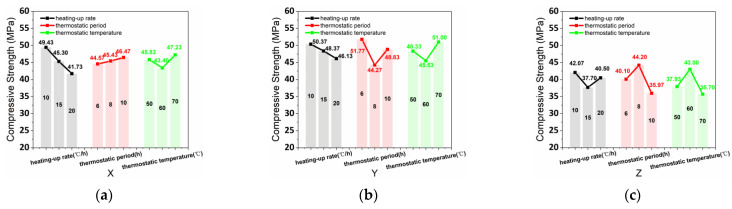
(**a**) CS_x_, (**b**) CS_y_, and (**c**) CS_z_ affected by various heating-up rates, thermostatic periods, and thermostatic temperatures.

**Figure 7 materials-15-02864-f007:**
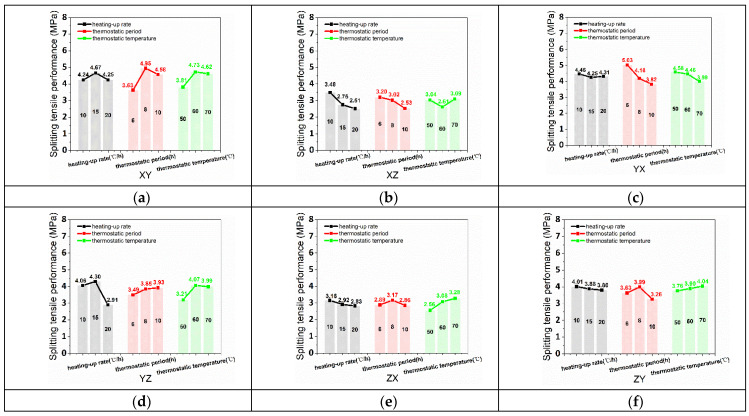
(**a**) TS_xy_, (**b**) TS_xz_, (**c**) TS_yx_, (**d**) TS_yz_, (**e**) TS_zx_, and (**f**) TS_zy_ affected by various heating-up rates, thermostatic periods, and thermostatic temperatures.

**Figure 8 materials-15-02864-f008:**
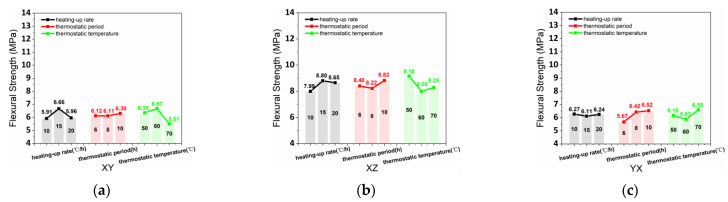
(**a**) FS_xy_, (**b**) FS_xz_, (**c**) FS_yx_, (**d**) FS_yz_, (**e**) FS_zx_, and (**f**) FS_zy_ affected by various heating-up rates, thermostatic periods, and thermostatic temperatures.

**Figure 9 materials-15-02864-f009:**
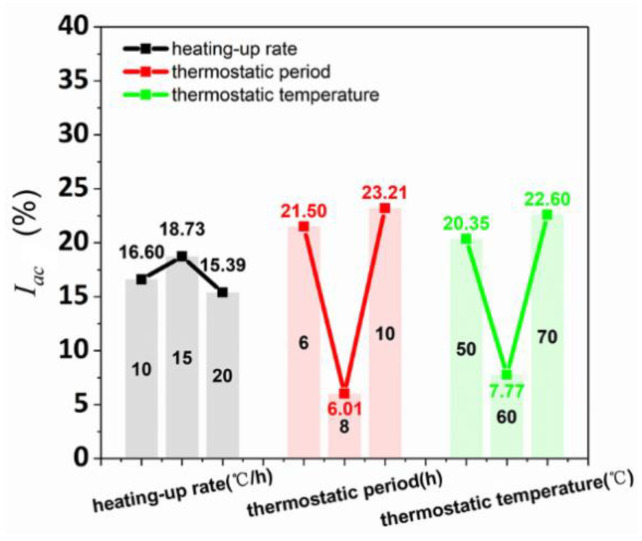
I_ac_ affected by various heating-up rates, thermostatic periods, and thermostatic temperatures.

**Figure 10 materials-15-02864-f010:**
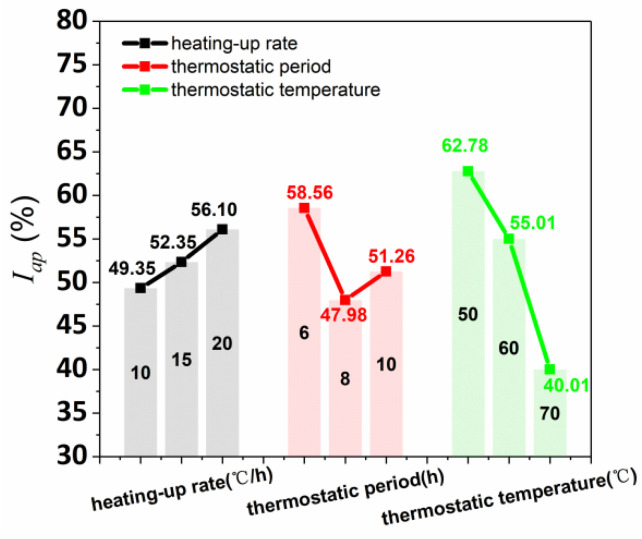
I_ap_ affected by various heating-up rates, thermostatic periods, and thermostatic temperatures.

**Figure 11 materials-15-02864-f011:**
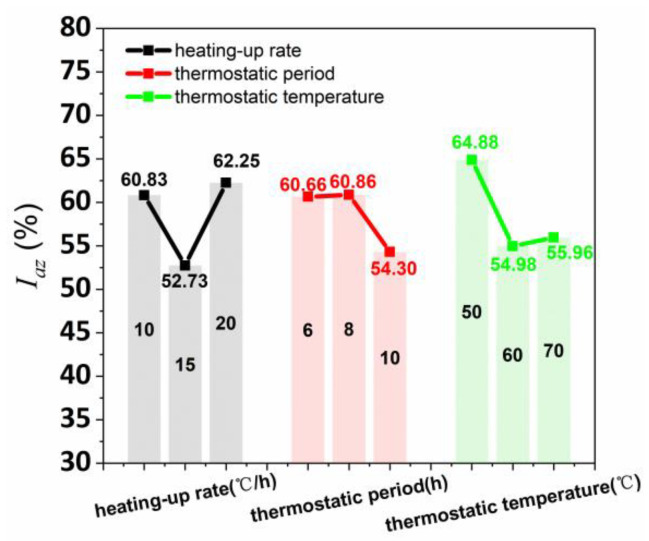
I_az_ affected by various heating-up rates, thermostatic periods, and thermostatic temperatures.

**Figure 12 materials-15-02864-f012:**
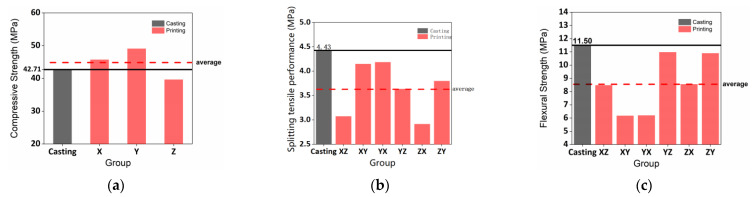
The mechanical performance average comparison of cast and printed samples in the varying directions: (**a**) compressive performance, (**b**) splitting tensile performance, (**c**) flexural performance.

**Figure 13 materials-15-02864-f013:**
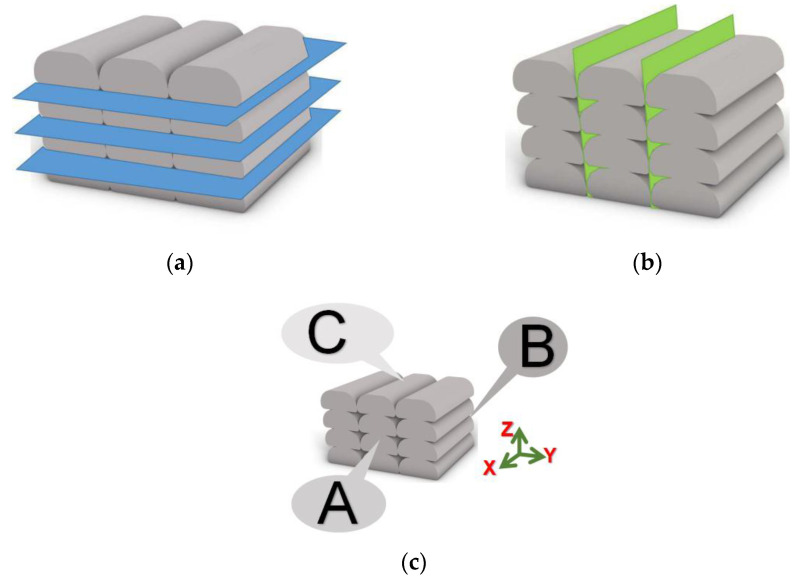
The horizontal interlayer diagrammatic sketch between printing layers (**a**), the vertical interlayer diagrammatic sketch between printing filaments (**b**), and the diagrammatic sketch of printed cube sample faces (**c**).

**Figure 14 materials-15-02864-f014:**
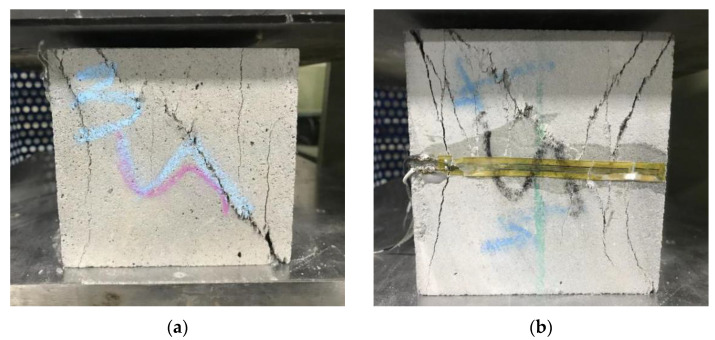
The printed cube sample cracks of (**a**) C and (**b**) reverse plane on the X-direction compression.

**Figure 15 materials-15-02864-f015:**
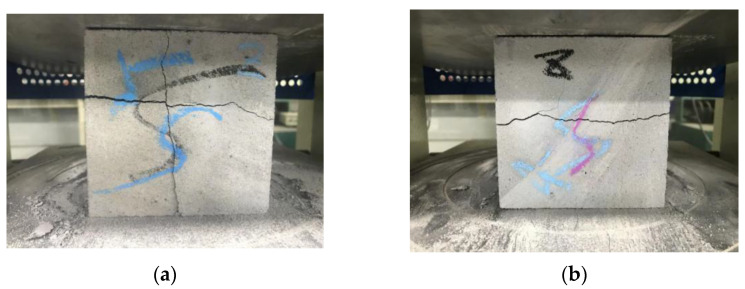
The printed cube sample cracks of (**a**) C plane and (**b**) reverse plane.

**Figure 16 materials-15-02864-f016:**
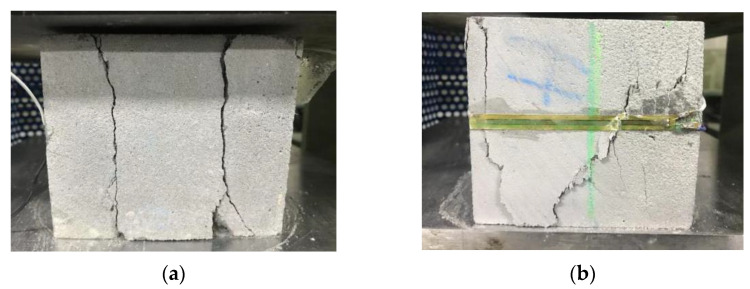
The printed cube sample cracks of (**a**) A and (**b**) reverse plane.

**Table 1 materials-15-02864-t001:** Chemical compositions, specific surface area, fineness of 45μm sieve residue, and density of the cement, FA, SF, and GGBS (wt.%).

Precursors	Content (wt.%)										Specific Surface Area (m^2^/kg)	Fineness of 45μm Sieve Residue (%)	Density (kg/m^3^)
	SiO_2_	Al_2_O_3_	Fe_2_O_3_	CaO	Ignition Loss	MgO	SO_3_	K_2_O	Na_2_O	Cl			
Cement	19.4–21.5	4.1–4.9	2.8–2.9	61.9–64.2	1.9–2.0	1.1–1.2	3.0–3.2	0.6–0.7	0.1–0.2	0.01–0.02	350–400	-	2800–3200
FA	43–54	28–34	8–13	0.4–0.5	1.6–4.7	1.1–2.3	0.5–1.2	2–4	0.8–1.5	0.01–0.02	-	6.7–7.3	2400–2500
SF	93–97	-	-	0.26–0.28	1.0–1.1	-	-	-	-	0.01–0.02	20,000–27,000	2.2–4.1	320–380
GGBS	34.7–38.2	9.1–10.2	0.5–0.7	38.8–40.5	0.6–0.8	9.9–11.1	0.1–1.8	0.12–0.14	0.24–0.29	0.01–0.04	420–480	5.8–7.5	2800–2900

**Table 2 materials-15-02864-t002:** Main properties of the PVA fibers.

Diameter (μm)	Length (mm)	Density (kg/m^3^)	Elastic Modulus (GPa)	Tensile Strength (MPa)	Elongation (%)
39	18	1.2	76.5	1950	6

**Table 3 materials-15-02864-t003:** Control mix proportion of this test.

Components (wt.%)								
Cement	FA	SF	GGBS	Sand	Water	Polycarboxylate Superplasticizer	PVA Fiber	Hydroxypropyl Methylcellulose
55	20	15	10	130	36	0.23	0.21	0.016

**Table 4 materials-15-02864-t004:** The curing conditions of samples in this test.

Test Group	Curing Conditions		
	e (°C/h)	E (h)	f (°C)
1	10	6	50
2	10	8	70
3	10	10	60
4	15	6	70
5	15	8	60
6	15	10	50
7	20	6	60
8	20	8	50
9	20	10	70
C_c_	Curing in natural condition		
C_p_			

**Note**: e represents heating-up rate; E represents thermostatic period; f represents thermostatic temperature, C_c_ represents control cast sample, C_p_ represents control printed sample, natural condition is 20 °C average temperature and 60% humidity.

**Table 5 materials-15-02864-t005:** Orthogonal experimental table of various test group designs, curing conditions, compressive strength test data, and coefficients of variation.

Test Group	Curing Conditions			Compressive Strength (MPa)			Coefficient of Variation (%)		
	e (°C/h)	E (h)	f (°C)	CS_x_	CS_y_	CS_z_	CS_x_	CS_y_	CS_z_
1	10	6	50	48.0	55.0	36.0	5.12	3.75	9.13
2	10	8	70	52.0	45.8	46.9	7.20	8.86	9.51
3	10	10	60	48.3	50.3	43.3	7.66	6.81	5.58
4	15	6	70	46.1	56.7	39.4	8.60	5.72	9.27
5	15	8	60	42.3	42.7	40.8	4.89	7.21	8.66
6	15	10	50	47.5	45.7	32.9	5.59	4.98	9.40
7	20	6	60	39.6	43.6	44.9	6.43	5.61	5.08
8	20	8	50	42.0	44.3	44.9	5.99	6.45	7.37
9	20	10	70	43.6	50.5	31.7	7.02	4.41	8.81
C_c_	Curing in natural condition			42.7			7.53		
C_p_				48.1	56.3	35.8	4.37	3.80	7.96

**Note**: CS_x_ represents compressive capacity in the X direction; CS_y_ represents compressive capacity in the Y direction; CS_z_ represents compressive capacity in the Z direction; C_c_ means control cast sample; C_p_ means control printed sample.

**Table 6 materials-15-02864-t006:** Orthogonal experimental table of various test group designs, curing conditions, and splitting tensile strength test data.

Test Group	Curing Conditions			Splitting Tensile Strength (MPa)					
	e (°C/h)	E (h)	f (°C)	TS_xy_	TS_xz_	TS_yx_	TS_yz_	TS_zx_	TS_zy_
1	10	6	50	2.680	4.201	5.468	3.081	2.788	4.253
2	10	8	70	5.380	3.488	4.113	4.520	3.807	4.087
3	10	10	60	4.654	2.756	3.813	4.591	2.896	3.693
4	15	6	70	4.024	3.164	4.348	4.316	2.891	4.902
5	15	8	60	5.367	2.865	4.278	4.546	3.336	3.801
6	15	10	50	4.628	2.228	4.138	4.049	2.547	2.947
7	20	6	60	4.182	2.222	5.284	3.081	2.999	4.208
8	20	8	50	4.113	2.699	4.145	2.489	2.355	4.074
9	20	10	70	4.456	2.611	3.501	3.145	3.145	3.126
C_c_	Curing in natural condition			4.43					
C_p_				4.533	2.018	2.814	2.588	2.426	2.909

**Note**: TS_xy_ represents splitting tensile strength in the XY direction; TS_xz_ represents splitting tensile strength in the XZ direction; TS_yx_ represents splitting tensile strength in the YX direction, TS_yz_ represents splitting tensile strength in the YZ direction, TS_zx_ represents splitting tensile strength in the ZX direction, TS_zy_ represents splitting tensile strength in the ZY direction.

**Table 7 materials-15-02864-t007:** Orthogonal experimental table of various test group designs, curing conditions, and variation coefficient of splitting tensile strength.

Test Group	Curing Conditions			Coefficient of Variation (%)					
	e (°C/h)	E (h)	f (°C)	TS_xy_	TS_xz_	TS_yx_	TS_yz_	TS_zx_	TS_zy_
1	10	6	50	2.34	5.27	5.90	9.27	1.93	3.99
2	10	8	70	9.46	7.40	10.10	7.56	5.17	5.60
3	10	10	60	4.43	8.34	7.93	2.45	6.50	2.39
4	15	6	70	8.60	9.12	9.25	6.06	9.78	7.08
5	15	8	60	10.03	5.59	9.74	4.64	4.37	2.65
6	15	10	50	5.43	3.45	8.65	5.73	8.08	3.93
7	20	6	60	9.81	7.06	7.51	1.48	5.59	2.64
8	20	8	50	7.77	2.68	5.15	4.84	6.83	8.89
9	20	10	70	9.91	3.93	10.78	9.07	8.66	6.75
C_c_	Curing in natural condition			4.15					
C_p_				5.34	3.78	9.71	5.39	7.13	1.85

**Table 8 materials-15-02864-t008:** Orthogonal experimental table of various test group designs, curing conditions, and flexural strength test data.

Test Group	Curing Conditions			Flexural Strength (MPa)					
	e (°C/h)	E (h)	f (°C)	TS_xy_	FS_xz_	FS_yx_	FS_yz_	FS_zx_	FS_zy_
1	10	6	50	5.94	8.10	6.12	12.06	8.28	11.70
2	10	8	70	5.65	7.56	6.74	10.29	8.64	11.65
3	10	10	60	6.14	8.31	5.94	10.64	8.98	10.60
4	15	6	70	5.55	8.98	5.64	9.59	7.17	9.46
5	15	8	60	7.00	7.57	6.43	11.08	9.91	10.38
6	15	10	50	7.42	9.85	6.25	11.09	7.94	11.74
7	20	6	60	6.87	8.11	5.25	10.37	10.29	11.16
8	20	8	50	5.68	9.53	6.10	12.45	8.29	12.37
9	20	10	70	5.34	8.31	7.36	11.29	7.57	9.05
C_c_	Curing in natural condition			11.50					
C_p_				5.63	9.57	6.13	12.72	8.97	12.20

**Note**: FS_xy_ represents flexural strength in the XY direction, FS_xz_ represents flexural strength in the XZ direction; FS_yx_ represents flexural strength in the YX direction, FS_yz_ represents flexural strength in the YZ direction, FS_zx_ represents flexural strength in the ZX direction, FS_zy_ represents flexural strength in the ZY direction.

**Table 9 materials-15-02864-t009:** Orthogonal experimental table of various test group designs, curing conditions, and variation coefficient of flexural strength.

Test Group	Curing Conditions			Coefficient of Variation (%)					
	e (°C/h)	E (h)	f (°C)	FS_xy_	FS_xz_	FS_yx_	FS_yz_	FS_zx_	FS_zy_
1	10	6	50	5.33	3.58	8.68	6.14	7.85	8.41
2	10	8	70	7.21	5.47	9.33	5.49	8.29	5.21
3	10	10	60	10.3	4.18	6.75	6.35	8.89	5.47
4	15	6	70	6.81	3.93	10.26	5.37	9.37	5.26
5	15	8	60	4.5	6.11	5.37	2.99	5.06	4.47
6	15	10	50	5.16	4.51	7.29	5.1	7.11	7.2
7	20	6	60	8.48	6.4	9.2	6.8	10.03	1.8
8	20	8	50	8.28	1.65	8.93	7.2	6.13	8.39
9	20	10	70	7.08	9.51	9.19	6.23	8.55	6.19
C_c_	Curing in natural condition			3.8					
C_p_				2.15	6.07	5.68	3.74	1.15	5.14

**Table 10 materials-15-02864-t010:** Orthogonal experimental table of various test group designs, curing conditions, and anisotropy coefficient of mechanical performance.

Test Group	Curing Conditions			Compressive Performance	Splitting Tensile Performance	Flexural Performance
	e (°C/h)	E (h)	f (°C)	I_ac28_ (%)	I_at_ (%)	I_af_ (%)
1	10	6	50	29.33	65.00	68.10
2	10	8	70	9.70	34.80	59.62
3	10	10	60	10.78	48.25	54.78
4	15	6	70	26.03	43.55	54.03
5	15	8	60	3.38	49.64	50.29
6	15	10	50	26.78	63.85	53.88
7	20	6	60	9.15	67.14	59.86
8	20	8	50	4.95	59.49	72.67
9	20	10	70	32.07	41.67	54.23
C_p_	Curing in natural condition			31.22	67.41	71.87

**Note**: I_ac28_ represents anisotropy coefficient of compressive performance, I_ap_ represents anisotropy coefficient of splitting tensile performance, and I_az_ represents anisotropy coefficient of flexural performance.

## Data Availability

The data presented in this study are openly available.

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
