# Peer review of "Mechanical Performance of 3D Printed Concrete in Steam Curing Conditions"

_materials, 2022, doi:10.3390/ma15082864_

Round 1
Reviewer 1 Report
In this study entitled "Mechanical Performance of 3D Printed Concrete in Steam Curing Conditions” the authors endeavored to investigate the effect of steam curing on mechanical properties of 3D printed composites containing various mineral admixtures as well as to optimize these steam curing parameters. However, what has been done to achieve this aim leaves a lot to be desired making this paper not suitable for publication in its current condition. Nonetheless, for the development of this article the following recommendations have to be considered by the authors.
- The language of this article is very poor, efforts have to be made to improve its quality especially when it comes to the selection of words. Proofreading should be considered.
- The introduction is not enough and should be developed, sufficient background information on steam curing and its effects must be provided. Question such as “why we feel the need for steam curing in 3DPC” should be answered.
- Authors should explain the reason of using quartz sand rather than normal sand since economical wise quartz sand is more expensive. Giving knowledge on what led them to use the other types of pozzolans particularly silica fume should be explained
- Material properties in table 1. are insufficient physical and chemical properties (One of the most important chemical compositions of pozzolans is their Al2O3, should be given) of these materials have to be provided.
- In table 1. What did the authors mean by the property “loss”?
- What were the properties of water reducer and Hydroxypropyl methylcellulose used in this study?
- It is stated that a high-efficiency polycarboxylic acid water reducer was used in the study, however in table 3. Water reducer is mentioned. Knowing the difference between a high range water reducing admixture and a water reducing admixture this should be checked and corrected.
- The mix proportion used in this study should be explained, why was this mix design selected?
- Based on what parameters was the mix designed presented in this study characterized as extrudable?
- Based on what criteria were the steam curing parameters selected?
- Which was the printing direction?
- A great deal of relevant information regarding the printing parameters has not been provided. Parameters such as open time, nozzle height etc. are known to be considerably effective on the properties of printed composites. These should be added and the reason of their selection defined.
- The aspect ratio of the nozzle used in this study is bigger than 2 which is different to that used in many other previous studies. This should be explained.
- In figure 4. Where the cutting of specimens is showed on images, based on these images it is clear that some specimens were cut to possess unequal number of interlayer (both from sides and above and below) while other were cut to only possess the below and above interlayer zones. Knowing the interlayer effect especially in 3DCP how did the Authors compare these two very distinct kinds of specimens?
- Line 148, what is it meant by natural maintenance?
Reviewer 2 Report
The paper focus on curing condition problem in 3D concrete / mortar printing. In my opinion this is important subject in 3D concrete / mortar printing technology. In addition Authors establish the anisotropy coefficient which could help to find the optimal curing condition. Despite important topic of this study the manuscript should be revised according to remarks below. The Authors missed some important information which could help to fully understand the manuscript.
Point 1. Maybe Authors should consider add some related research: 1) The different temperature curing was discussed in https://doi.org/10.1051/matecconf/201821903008 2) The curing condition including drying aspects was evaluated in https://doi.org/10.3390/ma13081800. This work also contain information about saving in costs and labor (presented in Your paper at lines 40 – 41).
Line 87 – “grading range between 40 mesh and 70 mesh” – the grading range should be described in SI unit (for particle distribution size “mm” unit are often used).
Point 2.1 – the Authors should add the particle size distribution curve for aggregate
Table 1. – Lack of information upon specific surface for FA and density for cement and FA.
Table 3 – and line 104 – 106 – I understood that Authors refer to binder in wt.% ration, but the Table 3 are very difficult to understand (in addition I see some mistakes i. e. sum of binder is equal to 101%). The Authors should show the mix composition in kg/m3 (for example Table 5 in https://doi.org/10.1016/j.jobe.2021.103030)
Point 2.2. and 2.3 – The Authors should explain the tests better: a) in Table 4 the group named Cc and Cp are tested after 28 days in 20°C?; b) the test groups 1 to 9 are exposed on different curing condition but: when the curing condition has started? (I mean when the heating – up period was stated, immediately after printing or later?); c) If the curing condition consist only one cycle or is it a loop?
Table 5, Fig. 6, Table 6, Fig. 7, Table 7, Fig. 8 – the Authors should add the some statistical values (Coefficient of variation or standard deviation) for results.
Round 2
Reviewer 1 Report
‘Therefore, research on the optimum steam curing conditions of mechanical capacity and anisotropy of 3DCP is urgent.’ "Required" instead of "urgent" would be better.
"Moreover, the addition of hydroxypropyl methylcellulose improved flowability." please reconsider this. HPMC cannot increase flowability, but can inrease the viscosity.
"the rectangular nozzle can improve effective contact area between printing filaments at the deposition process of materials." This should be supported by previous studies.
Reviewer 2 Report
The Authors revised manuscript according to my remarks.
In addition the Authors should check the copyright for previously published in [1] figures: Fig.1, Fig. 3, Fig. 4b). Maybe the reference to paper [1] should be added.
[1] https://doi.org/10.3390/buildings12030374
